# Selective Oxidation of Transient Organic Radicals in the Presence of Gold Nanoparticles

**DOI:** 10.3390/nano11030727

**Published:** 2021-03-14

**Authors:** Viacheslav Shcherbakov, Sergey A. Denisov, Mehran Mostafavi

**Affiliations:** Institute de Chimie Physique (ICP), CNRS/Université Paris-Saclay, 91405 Orsay, France; viacheslav.shcherbakov@universite-paris-saclay.fr

**Keywords:** gold nanoparticle, radical, ROS, catalysis, oxidation, radiolysis, radiosensitization

## Abstract

The ability of gold nanoparticles (AuNPs) to catalyze reactions involving radicals is poorly studied. However, AuNPs are used in applications where chemical reactions involving transient radicals occur. Herein, we investigate AuNPs’ catalytic effect on 2-propanol oxidation and acetanilide hydroxylation in aqueous solutions under ionizing radiation at room temperature. In both cases, the presence of AuNPs led to selective oxidation of organic radicals, significantly changing the products’ composition and ratio. Based on these observations, we stress how AuNPs’ catalytic activity can affect the correctness of reactive oxygen species concentration determination utilizing organic dyes. We also provide a discussion on the role of AuNPs’ catalytic activity in the radiosensitization effect actively studied for radiotherapy.

## 1. Introduction

For a long time, gold was not considered a promising catalyzer. In early studies, gold demonstrated less catalytic activity for hydrogenation and dehydrogenation than Pt and Pd [1,2]. However, 15 years later, it was found that small gold particles on α-Fe_2_O_3_, Co_3_O_4_, or NiO supports efficiently catalyze CO oxidation at low temperatures [3]. This finding provoked great interest in studying the catalytic properties of gold nanoparticles (AuNPs) [4]. The industrial search for promising catalyzers led to a bias in research towards supported AuNPs compared to gold colloids [5].

For the first time, the catalytic activity of AuNPs under normal conditions was demonstrated by radiation chemistry and electrochemistry communities in the 1980s [6,7]. In electrochemistry, gold and other conductive nanoparticles, e.g., silver, have been studied to catalyze redox reactions [8,9], while in radiation chemistry, such properties were used mainly to increase hydrogen production in aqueous solutions of alcohols under radiation [10,11,12]. Nonetheless, these works went unnoticed. Even in radiation chemistry, there are only a few examples studying the interaction of AuNPs with transient organic radicals not devoted to H_2_ production [13,14,15]. For example, nitroxyl-free radicals’ interaction with AuNPs was studied by electron paramagnetic resonance spectroscopy, which confirmed the interaction of unpaired electrons with the AuNP’s surface [16]. Lien et al. [17], using a 5-tert-butoxycarbonyl-5-methyl-1-pyrroline N-oxide (BMPO) spin trap for ^•^OH radicals, showed that AuNPs catalyze the conversion of ^•^BMPO–OH radical to ^•^BMPO–H radical.

Understanding AuNPs’ catalytic effect is essential for biomedical applications [18,19,20], where these particles are still often considered an inert material, such as bulk gold [21]. Considering the findings on catalytic properties, one can assume that AuNPs could alter cell metabolism by oxidizing organic radicals naturally produced in the cell due to the presence of reactive oxygen species (ROS) such as hydroxyl radical (^•^OH), superoxide radical (O_2_^•−^), and hydrogen peroxide (H_2_O_2_).

Despite the pioneering works utilizing AuNPs as catalysts in radiation chemistry, their catalytic activity is neglected in radiotherapy research, where they represent an essential class of radiosensitizing agents—materials that increase radiation damage [22]. The radiosensitizing effect was initially explained by a physical dose enhancement due to the higher mass energy-absorption coefficient of gold than water/soft tissues [23,24]. Nowadays, it is clear that the effect is much more complex and involves different physical, chemical, and biological processes [24,25,26,27,28]. The catalytic activity of AuNPs in radiosensitization studies is mainly mentioned in the context of ROS production, which is claimed as one of the mechanisms of AuNPs’ radiosensitizing effect [25,27]. However, in recent work, we showed that at a therapeutically relevant gold concentration (<3 mM atomic gold, <600 µg cm^−3^), the presence of gold nanoparticles does not induce higher primary radicals formation [29].

It is important to note that ROS are generally measured by an indirect method using fluorescence spectroscopy with organic dyes [27]. This method is based on converting a non-fluorescence molecule into a fluorescent one in reaction with ROS [30,31]. The amount of formed fluorescence product is claimed to be proportional to ROS concentration. However, the mechanism of such oxidation is complex. The oxidation always occurs through the formation of transient radicals. Therefore, not considering the catalytic activity of AuNPs towards transient organic radicals can lead to incorrect determination of ROS concentration.

In this work, we focus on the oxidation of transient organic radicals in the presence of AuNPs. We have studied two types of organic radicals produced by ^•^OH radical attack: H atom abstraction from 2-propanol and ^•^OH attachment to a benzene ring of acetanilide. In both cases, the presence of AuNPs led to selective oxidation of organic radicals. By extrapolating our results, we discuss AuNPs’ catalytic activity’s effect on ROS detection by fluorescent dyes, its potential role in explaining the radiosensitizing effect of AuNPs reported in the literature, and general precautions of utilizing AuNPs in biomedical applications.

## 2. Materials and Methods

### 2.1. Chemicals

Gold nanoparticle synthesis: potassium gold (III) chloride (98%), sodium borohydride (98%), and sodium citrate (99%) were purchased from Sigma Aldrich (St. Quentin Fallavier, France). Sodium chloride (99.9%) was purchased from VWR chemicals (VWR International Inc., Fontenay-sous-Bois, France), and 2-propanol (99.5%, Sigma Aldrich, St. Quentin Fallavier, France) was used as a hydroxyl radical scavenger. Acetone (99.8%, Sigma Aldrich, St. Quentin Fallavier, France) and N_2_O (Alpha Gaz, Air Liquide, Paris, France) were used as hydrated electron scavengers. Ar (Alpha Gaz, Air Liquide, Paris, France) was used for deoxygenation. Acetanilide (≤100%, Jeulin, Évreux, France) was used as a model for hydroxylation reaction. Furthermore, 2-Hydroxyacetanilide (97%) and 3-Hydroxyacetanilide (99+%) purchased from Acros Organics (Fisher Scientific, Illkirch, France) and 4-Hydroxyacetanilide (>99%) purchased from Sigma-Aldrich (St. Quentin Fallavier, France) were used as standards for HPLC. All chemicals were used without additional purification. Water (Milli-Q, Merck KGaA, Darmstadt, Germany) with a resistivity of 18 MΩcm was used in all experiments.

### 2.2. Synthesis of AuNPs

AuNPs were prepared by reduction with sodium borohydride: 3 mL of 100 mM stock solution of gold salt was mixed with 87.7 mL of deionized water, and 9.3 mL of 100 mM NaBH_4_ was added to the solution while stirring for 15 min. The solution changed color from yellow to dark red on the reduction of gold ions. The final concentration of gold salt was 3 mM. The particles were stable for weeks. Their zeta potential was −20 mV (Zetasizer Nano ZS, Malvern, Palaiseau, France). The maximum wavelength of the plasmon band was 520 nm. 

The diameters of nanoparticles were determined from the maximum wavelength of the AuNPs’ plasmon band (520 nm) [32] and theirs size (20 nm) was confirmed by transmission electron microscopy (STEM 1400, JEOL Ltd., Tokio, Japan). The pH of the solutions was ~7. All experiments were conducted at 22.5 °C.

The concentration of particles was calculated as follows:$$C_{\mathrm{AuNP}}=\frac{6\;M_{w}\;C}{\pi \rho d^{3}N_{A}}$$
where *M_w_*: gold molecular weight (197 g/mol); *C*: molar atomic gold concentration; ρ: density of bulk gold (19.32 g cm^−3^); N_*A*_: Avogadro number; and *d*: diameter of particles.

### 2.3. Gamma Radiolysis

A ^60^Co γ source was used for steady-state radiolysis experiments. The dose rate was verified by Frick dosimetry. No dose rate dependence was observed in the used dose rate ranges in both experiments. Samples were irradiated in glass bottles. AuNPs were precipitated by the addition of NaCl up to 0.5 wt.% and centrifugation utilizing an Eppendorf MiniSpin centrifuge (Eppendorf Inc., Montesson, France) with F-45-12-11 rotor at a speed of 13,400 rpm for 30 min. The samples were placed in 2-mL Eppendorf microtubes.

### 2.4. 2-Propanol Oxidation

Briefly, 4 mL of an aqueous solution containing 100 mM of 2-propanol and 20 mM of acetone and AuNP suspension (12 nM particle concentration) with the same amount of 2-propanol and acetone were irradiated at various doses. The dose rate was in the range between 0.2 and 4 kGy h^−1^. All samples (4 mL) were deoxygenated by bubbling with Ar for 10 min. The acetone concentration formed during irradiation was determined by the acetone absorbance band at 265 nm using UV–Vis absorption spectrophotometry (Hewlett Packard 8453 spectrophotometer, Hewlett Packard Inc., Puteaux, France). The extinction coefficient of 15.4 M^−1^ cm^−1^ determined in the laboratory is within the literature data [33]. 

### 2.5. Acetanilide Hydroxylation 

Briefly, 2 mL of 0.5 mM acetanilide aqueous solution in the absence and presence of 6 nM AuNPs was irradiated at a dose rate in the range of 20 to 70 Gy min^−1^. All samples (2 ml) were saturated with N_2_O by bubbling for 10 min. Irradiated samples were analyzed by high-performance liquid chromatography (Agilent Technologies 1260 infinity, Agilent Technologies, Les Ulis, France). A diode-array detector was used to record the light intensity in the range from 200 to 500 nm. The HPLC column was EC 50/4.6 Nucleoshell RP 18plus, 2.7 µm. A mixture of 97% water and 3% acetonitrile was used for isocratic elution at a rate of 1.5 mL min^−1^. The products of acetanilide hydroxylation were determined by comparing their spectra and retention time with commercial standards.

## 3. Results

The ^•^OH radicals were produced by water radiolysis under gamma radiation (^60^Co). The radiolytic yields (G, mol J^−1^) of primary water radicals and molecules are well established and represent the number of molecules produced per joule of energy. We have used standard values for the homogenous step of water radiolysis (1); G values (×10^−7^mol J^−1^) are given in brackets [34].
(1)H2O→γ eaq−(2.8), H3O+, H •(0.6), O •H(2.8), H2O2 (0.7)

All experiments were carried out with AuNPs of 20 nm in diameter, synthesized using sodium borohydride as a reduction agent (see Section 2 and Appendix A). The suspensions with a concentration of gold atoms of 3 and 1.5 mM correspond to 12 and 6 nM of 20-nm particles, respectively. Pure water was used as a reference. A series of experiments were conducted to prove that water and supernatant show similar results (Appendix A), which proved that water could be considered an appropriate reference.

### 3.1. Oxidation of Organic Radical Produced by ^•^H Atom Abstraction

As mentioned in the Introduction, metal nanoparticles have been shown to increase hydrogen production in irradiated aqueous solutions of aliphatic alcohols [10,11,12]. This phenomenon was explained by the catalytic reduction in water on the metal surface. However, another side of the process, namely the oxidation of organic radicals which work as electron donors, has never been studied. 

Therefore, we first investigated the AuNPs’ effect on 2-propanol oxidation following acetone formation in an irradiated aqueous solution containing 100 mM of 2-propanol and 20 mM of acetone saturated with Ar. 

The acetone concentration was determined by optical absorption measurements at 265 nm using an extinction coefficient of 15.4 M^−1^ cm^−1^ (Appendix A) in the supernatants of AuNP suspensions (see Section 2 for details). The acetone’s radiolytic formation yield was determined from acetone concentration dependence versus the applied dose (Figure 1). In this experiment, the applied dose was relatively high (up to 220 kGy) to accumulate a measurable quantity of acetone.

The first step of 2-propanol (ROH) oxidation in irradiated aqueous solution is a hydrogen atom abstraction, mainly from the alpha carbon, by ^•^OH and ^•^H radicals, forming α-hydroxyisopropyl radical (R^•^OH) (Scheme 1 reactions 1–2) [35]. In the presence of acetone (20 mM), hydrated electrons (e^−^_aq_) are entirely scavenged by acetone, forming the same R^•^OH radical (Scheme 1 reaction 3).

Based on these reactions (Scheme 1), the maximum radiolytic yield of R^•^OH can be calculated according to the equation: G(R•OH) = G(eaq−) + G(O •H) + G(H•) = 6.2 × 10−7 mol J−1. The R^•^OH concentration can be calculated by multiplication of the radiolytic yield by dose (Figure 1, top axis).

The oxidation of the R^•^OH radical leads to acetone formation. We found that in the presence of AuNPs (12 nM of 20 nm), acetone formation is 3.5 times higher than in water (Figure 1), which corresponds to 59% and 17% of the maximum yield, respectively. 

This difference can be explained by direct oxidation of the R^•^OH radical in the presence of AuNPs, instead of a well-known disproportionation reaction occurring between two R^•^OH radicals, leading to acetone formation.

The R^•^OH oxidation in the presence of AuNPs suggests an electron transfer from the radical to the particle, as reported previously [8,36]. Conducting nanoparticles can accumulate a negative charge by uptaking unpaired electrons from the α-hydroxyisopropyl radical [36], which is supported by the concept of “floating electrocatalysis” [8]. The excess charge on the particle can be taken by electron acceptors, such as H^+^ or H_2_O in deaerated solutions, to fulfill the requirement of electroneutrality. AuNPs work as an electron relay conducting electron from a donor to an acceptor [37].

### 3.2. Oxidation of Organic Radicals Produced by ^•^OH Attachment

The ^•^OH radical can not only cause H-abstraction, but it also could form an ^•^OH adduct. Therefore, to study the catalytic effect of AuNPs on such a type of reaction, we chose acetanilide hydroxylation by ^•^OH, which is similar to other aromatic systems. It includes the following steps: attachment of ^•^OH to a benzene ring forming an adduct and the loss of an electron, followed by the loss of H^+^, forming a stable hydroxylated molecule (Scheme 2) [38,39]. In this reaction, numerous products are formed. Herein, we focused on three main products: hydroxyacetanilide with a hydroxyl group in ortho, meta, para positions, and acetanilide consumption (Figure 2).

Aqueous solutions of 0.5 mM acetanilide with and without 6 nM of AuNPs were irradiated (^60^Co) under a N_2_O atmosphere to convert all hydrated electrons to ^•^OH radicals (Appendix A).

The first observation consists of a drastic decrease in the number of products in the presence of AuNPs. Products numbered from 1 to 5 in Figure 2 are not detected in the presence of AuNPs. Secondly, acetanilide consumption is only slightly different for samples with and without AuNPs (Figure 3a). In the absence of AuNPs, the yield of acetanilide disappearance (5.0 × 10^−7^ mol J^−1^) is lower than the yield of ^•^OH radicals (5.6 × 10^−7^ mol J^−1^, Appendix A). The reason for that is likely the disproportionation reaction between two adducts that re-form acetanilide. In the presence of AuNPs, the absolute value of acetanilide disappearance is equal to the yield of ^•^OH radicals. It means that direct oxidation of acetanilide adducts by AuNPs dominates over the disproportionation reaction. Moreover, this result indicates that AuNPs do not induce higher ^•^OH radical production in contrast to the previously reported statement [27].

The total hydroxylation yield was increased four times from 20% to 79% (Table 1) in the presence of AuNPs (Figure 3b). Similar behavior was observed for phenylalanine oxidation under ionizing radiation in the presence of Fe(CN)_6_^−3^, leading to an increase in the formation yield of ortho-, meta-, and para-Tyrosine to 80% [39]. The remaining 20% was assumed to ^•^OH radical attachment to ipso position and ^•^H atom abstraction from the benzyl position, which do not lead to tyrosine formation. 

As in the case of 2-propanol, the radicals of acetanilide were selectively oxidized to hydroxyacetanilide in the presence of AuNPs.

## 4. Discussion

The usage of AuNPs in biomedical applications includes delivery, visualization, and radiosensitization [19,20,25]. For a long time, AuNPs were considered not only inert material but also not toxic. However, works appeared in the literature showing that gold nanoparticles do show toxic effects on cells [21]. 

Whenever the subject of AuNPs’ toxicity is raised, the explanation is higher ROS production in their presence with and without ionizing radiation, based on ROS measurement using the fluorescence spectroscopy method [40,41,42,43]. This method is based on the change in the fluorescence properties of dye when it reacts with ROS such as hydroxyl radical (^•^OH), superoxide radical (O_2_^•−^), and hydrogen peroxide (H_2_O_2_) [31]. For example, the detection of ^•^OH radicals is often based on a non-fluorescent probe molecule’s hydroxylation, forming a fluorescent one [30,44]. The reaction scheme is similar to the acetanilide hydroxylation described above: the formation of a transient radical (adduct) followed by its oxidation, forming a hydroxylated molecule. Thus, disregarding important AuNP catalytic properties may lead to interpretation of the increased formation of hydroxylation products as ^•^OH overproduction [45,46,47] since the product’s concentration is assumed to be proportional to ^•^OH concentration. Thus, detection of probe molecule consumption is a more appropriate indicator for ^•^OH detection. As we show here, the presence of AuNPs does not cause ^•^OH overproduction, in agreement with our previous work [29].

The extrapolation of our results towards fluorescent dyes used to detect ROS other than ^•^OH in the presence of metal nanoparticles is less obvious but still reasonable, since oxidation is a two-electron transfer process which occurs through the formation of transient organic radicals. Those radicals can be catalytically oxidized in the presence of nanoparticles, causing misdetermination of ROS production [27]. Thus, the validity of utilizing fluorescent probes without considering the catalytic effect of AuNPs in the presence of metal nanoparticles is under question.

Our results have significant consequences on the understanding of the mechanism of AuNPs as radiosensitization. Nowadays, it is considered to occur through AuNPs’ effect on different physical, chemical, and biological processes [24,25,26,27,28]. The radiosensitization mechanism is often discussed in the context of ROS overproduction in the presence of AuNPs [25,27]. However, AuNPs’ catalytic properties towards radicals were never discussed before.

It is essential to mention that different organic radicals are always present in cells due to their metabolism. Therefore, AuNPs can change the cell chemistry affecting radicals’ processes, even before ionizing radiation is applied, which, in principle, could shatter the cell’s resistivity towards ionization radiation.

## 5. Conclusions

We studied AuNPs’ catalytic activity towards the oxidation of two types of transient organic radicals produced by ^•^OH radical, namely by ^•^H atom abstraction from alpha carbon of 2-propanol, forming α-hydroxyisopropyl radical, and ^•^OH radical attachment to the benzene ring of acetanilide, forming different adducts. In both cases, AuNPs caused selective oxidation of transient organic radicals. Thus, AuNPs increased acetone formation by 3.5 times in an irradiated aqueous solution of 2-propanol/acetone. In the case of acetanilide hydroxylation, the number of products was decreased in the presence of AuNPs, and the total hydroxylation was increased by four times. At the same time, the consumption of acetanilide was equal to the concentration of ^•^OH produced during water radiolysis. This result contradicts those previously reported in the literature of ^•^OH overproduction in the presence of AuNPs.

We also discussed the implication of our results in the explanation of the radiosensitizing effect of AuNPs. We proposed that AuNPs can affect cell chemistry by catalytic oxidation of organic radicals, which can be produced naturally or by ionizing radiation.

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
