# Peer review of "Selective Oxidation of Transient Organic Radicals in the Presence of Gold Nanoparticles"

_nanomaterials, 2021, doi:10.3390/nano11030727_

Round 1

Reviewer 1 Report

The manuscript by Shcherbakov et al describes the study of gold nanoparticles behaviour in the radiolytic oxidation of two model substrates. The manuscript, in general, is accurate and well written. I agree with the Authors’ conclusion about the absence of overproduction of *OH radicals catalysed by gold nanoparticles (lines 237-238). On the other hand, there are plenty of problematic statements and descriptions in the manuscript, which must be corrected before the manuscript can be recommended for publication.

1) Please specify the gold salt kind (chloride, I suspect) right in the synthetic protocol 2.2 (line 86). Also, please specify reagents grade (e.g. >99%), at least in general.

2) There is a lack of definitions in the main text, e.g. line 25: AuNP; line 36: ERS; line 38: BMPO.

3) Lines 88-89: “The final concentration of gold salt was 3 mM.”. Why? How was it determined?

4) Line 91-92: “The diameters of nanoparticles were determined from the maximum wavelength of the AuNPs plasmon band (520 nm).” – please provide the respective literature reference.

5) Line 89: “Their zeta potential was -20 mV.” How was it determined?

6) Line 106: “4 ml of 100 mM of 2-propanol, and 20 mM of acetone aqueous solutions...” – Was it 4 mL of solution containing 100 mM of 2-propanol and 20 mM of acetone or two different solutions of 2-propanol and acetone, 4 mL each? Please clarify and correct this phrase.

7) Line 108: “All samples were saturated with Ar.”  - How was it performed? If Ar was bubbled through the mixture, the bubbling time must be specified. And if so, bubbling may reduce concentration of acetone, as it is a volatile compound.

8) Lines 110-111: “The extinction 110 coefficient of 15.4 M-1 cm-1 was used...” – Please provide the respective literature reference (or state that this coefficient was obtained by calibration if it was the case).

9) Lines 114-115: “Samples were 114 degassed with N2O.” – The same question as for Ar (see above). Please specify how degassing was performed.

10) Lines 143-144: “The acetone concentration was determined by optical absorption measurements at 143 265 nm using an extinction coefficient of 15.4 M-1 cm-1 (Figure S3)...”. Using the A=εcL equation one may find that the expected absorption of 20 mM of acetone for 1 cm cell (as stated in Figure S3) should be 0.31. However, it is clearly seen that absorption for 0 kGy spectrum in Figure S3 is considerably lower, being ca. 0.28. Please explain this difference and, if necessary, introduce respective corrections.

11) Line 161: “The oxidation of R•OH radical leads to acetone formation.” – This statement requires some comments. What is the mechanism of this oxidation (i.e. what is the oxidant here)? The Authors write further that “...a well-known disproportionation reaction occurring between two R•OH radicals, leading to acetone formation.” (lines 165-166). However, given the absence of air oxygen, this oxidation could also occur through the interaction of R*OH radical with *OH one (see, for instance, J. Phys. Chem. A, 2007, 111, 7736), what influences the radiolytic yields as the formation of acetone needs now two *OH radicals, but not one (as shown in Scheme I). Please check carefully and correct all the quantum (radiolytic) yields, if necessary. Please also evaluate the oxidation pathways of 2-propanol radical. The same problem concerns oxidation of acetanilide (Scheme II).

Author Response

REVIEWER 1

The manuscript by Shcherbakov et al describes the study of gold nanoparticles behaviour in the radiolytic oxidation of two model substrates. The manuscript, in general, is accurate and well written. I agree with the Authors’ conclusion about the absence of overproduction of *OH radicals catalysed by gold nanoparticles (lines 237-238). On the other hand, there are plenty of problematic statements and descriptions in the manuscript, which must be corrected before the manuscript can be recommended for publication.

ANSWERS:

We thank the reviewer for his/her comments and evaluation of our work.

Below we will reply comment by comment.

1. Please specify the gold salt kind (chloride, I suspect) right in the synthetic protocol 2.2 (line 86). Also, please specify reagents grade (e.g. >99%), at least in general.

The type of gold salt was written in part 2.1. Chemicals, line 75 “potassium gold (III) chloride (98%)”. We have also added grades of all used chemicals.

2. There is a lack of definitions in the main text, e.g. line 25: AuNP; line 36: ERS; line 38: BMPO.

The term gold nanoparticles and its abbreviation “AuNPs” was first mentioned in the abstract (line 9). However, we wrote it again in the introduction part “catalytic properties of gold nanoparticles (AuNPs)” (line 25-26).

ERS was changed to the full name of the method “studied by electron paramagnetic resonance spectroscopy” (line 36).

BMPO is a standard spin trap for OH radicals; in the literature it used as it is. We added the following text “using BMPO spin trap for •OH radicals” (line 38)

3. Lines 88-89: “The final concentration of gold salt was 3 mM.”. Why? How was it determined?

We selected this concentration to have the highest possible concentration of gold nanoparticles without any surfactant utilization. The same particles and concentration were also used in our previous paper [Shcherbakov, V., Denisov, S. A., & Mostafavi, M. (2020). On the Primary Water Radicals’ Production in the Presence of Gold Nanoparticles: Electron Pulse Radiolysis Study. Nanomaterials, 10, 2478 10.3390/nano10122478]. The gold salt concentration was verified by accurate measurement of the weight used for stock solutions preparation, which was also verified by gold ions electronic absorption at 220 nm.

4. Line 91-92: “The diameters of nanoparticles were determined from the maximum wavelength of the AuNPs plasmon band (520 nm).” – please provide the respective literature reference.

Reference #32 was added.

5. Line 89: “Their zeta potential was -20 mV.” How was it determined?

The zeta potential of AuNP was determined using “(Zetasizer Nano ZS, Malvern)”. Corresponding information was added to line 91.

6. Line 106: “4 ml of 100 mM of 2-propanol, and 20 mM of acetone aqueous solutions...” – Was it 4 mL of a solution containing 100 mM of 2-propanol and 20 mM of acetone or two different solutions of 2-propanol and acetone, 4 mL each? Please clarify and correct this phrase.

It was changed to:

Line 108: “4 ml of an aqueous solution containing 100 mM of 2-propanol and 20 mM of acetone, and AuNPs’ suspension (12 nM particle concentration) with the same amount of 2-propanol and acetone were irradiated at various doses.”

7. Line 108: “All samples were saturated with Ar.”  - How was it performed? If Ar was bubbled through the mixture, the bubbling time must be specified. And if so, bubbling may reduce the concentration of acetone, as it is a volatile compound.

Line 111: “All samples were deoxygenated by bubbling with Ar for 10 min.”

8. Lines 110-111: “The extinction 110 coefficient of 15.4 M-1 cm-1 was used...” – Please provide the respective literature reference (or state that this coefficient was obtained by calibration if it was the case).

Line 115: “The extinction coefficient of 15.4 M-1 cm-1 was determined in the laboratory that is lying within the literature data [33].”

9. Lines 114-115: “Samples were 114 degassed with N2O.” – The same question as for Ar (see above). Please specify how degassing was performed.

Line 118: “All samples were saturated with N2O by bubbling for 10 min“

10. Lines 143-144: “The acetone concentration was determined by optical absorption measurements at 143 265 nm using an extinction coefficient of 15.4 M-1 cm-1 (Figure S3)...”. Using the A=εcL equation one may find that the expected absorption of 20 mM of acetone for 1 cm cell (as stated in Figure S3) should be 0.31. However, it is clearly seen that absorption for 0 kGy spectrum in Figure S3 is considerably lower, being ca. 0.28. Please explain this difference and, if necessary, introduce respective corrections.

Indeed, acetone concentration in a non-irradiated solution (Figure S3) is 18.2 mM, based on the extinction coefficient of 15.4 M-1 cm-1. However, it is not an issue because, in this experiment, we measured acetone produced during radiation which equals to a difference between the initial concentration (before irradiation) and the final one. The initial amount of acetone was used to scavenge solvated electrons. Therefore, there is no difference between 18 or 20 mM, anyway, all solvated electrons will be scavenged. Of course, in each set of experiments, we use an actual initial acetone concentration, which was around 20 mM.

11. Line 161: “The oxidation of R•OH radical leads to acetone formation.” – This statement requires some comments. What is the mechanism of this oxidation (i.e., what is the oxidant here)? The Authors write further that “...a well-known disproportionation reaction occurring between two R•OH radicals, leading to acetone formation.” (lines 165-166). However, given the absence of air oxygen, this oxidation could also occur through the interaction of R*OH radical with *OH one (see, for instance, J. Phys. Chem. A, 2007, 111, 7736), what influences the radiolytic yields as the formation of acetone needs now two *OH radicals, but not one (as shown in Scheme I). Please check carefully and correct all the quantum (radiolytic) yields, if necessary. Please also evaluate the oxidation pathways of 2-propanol radical. The same problem concerns the oxidation of acetanilide (Scheme II).

By the statement on line 161, “The oxidation of R•OH radical leads to acetone formation.” we mean that if R•OH radical loses one electron, it will form acetone. Then we discuss that the main way of acetone formation in the absence of AuNPs is a disproportionation reaction. Further, we discuss that AuNP will eventually give the electron from R•OH to water or H+.

The stationary concentration of OH radicals and R•OH is low (10-9M) compared to the concentration of 2-propanol (100mM). Therefore, the reaction between OH radicals and 2-propanol is much more favorable than between OH radicals and R•OH

Reviewer 2 Report

This excellent paper reports on the investigation of catalytic properties of gold nanoparticles (AuNPs). The Authors have studied the oxidation of 2-propanol and hydroxylation of acetanilide in aqueous solutions under ionizing radiation in the presence of AuNPs. The Authors have shown that the formation of acetone by oxidation of α-hydroxyisopropyl radical (R*OH) in the presence of AuNPs is 3.5 times higher than without AuNPs. The Authors have also found that the total hydroxylation yield of acetanilide increases from 20% to 79% in the presence of AuNPs. The elucidation of the mechanism of the radiosensitizing effect of AuNPs has been presented. The discovery, reported in this paper, that AuNPs act as the radiosensitizers of certain cellular processes has significant consequences relating to understanding of cell’s resistance to ionization radiation. On one hand, this discovery is a warning of increased damages caused by ionizing radiation in the presence of AuNPs (and perhaps also other plasmonic and possibly semiconducting nanoparticles), and on the other hand, it may create new opportunities in applications of these effects in nanomedicine, e.g., for cancer treatment.  

I recommend the paper for publication after minor revision addressing the issues listed below:

  1. In Figures S1 C and S3, the ordinate variables should be followed only by units, e.g., “Absorbance, a.u.”, with explanation in Figure caption that 2 mm pathway cells were used. Alternatively, the correct mathematical expression: “A/5” (rather than “Absorption/5”), followed by “a.u.” units could be used.
  2. Figure captions, in main text and supporting information, should contain all important experimental conditions (e.g., concentrations of analytes, solutions, pH). In Figure 2 caption, the name of compound should be added.
  3. It seems that in addition to the radiosensitizing effects of AuNPs on the oxidation processes of certain small biomolecules, investigated by the Authors, AuNPs may also induce significant damage to DNA in the presence of ionizing radiation. For instance, it has been found that premutagenic changes in DNA are made by *OH and Cu(I)-hydroperoxyl complex radicals generated by Fenton-type mechanisms in the presence of Fe2+ or Cu2+ ion and catechol moieties (Mutation Research - Fundamental and Molecular Mechanisms of Mutagenesis, 2012, 735, 1– 11). Any comment on this subject would benefit general Readership.
  4. I have not found any typographical or English errors.

Author Response

REVIEWER 2

This excellent paper reports on the investigation of catalytic properties of gold nanoparticles (AuNPs). The Authors have studied the oxidation of 2-propanol and hydroxylation of acetanilide in aqueous solutions under ionizing radiation in the presence of AuNPs. The Authors have shown that the formation of acetone by oxidation of α-hydroxyisopropyl radical (R*OH) in the presence of AuNPs is 3.5 times higher than without AuNPs. The Authors have also found that the total hydroxylation yield of acetanilide increases from 20% to 79% in the presence of AuNPs. The elucidation of the mechanism of the radiosensitizing effect of AuNPs has been presented. The discovery, reported in this paper, that AuNPs act as the radiosensitizers of certain cellular processes has significant consequences relating to the understanding of cell’s resistance to ionization radiation. On the one hand, this discovery is a warning of increased damages caused by ionizing radiation in the presence of AuNPs (and perhaps also other plasmonic and possibly semiconducting nanoparticles), and on the other hand, it may create new opportunities in applications of these effects in nanomedicine, e.g., for cancer treatment.  

I recommend the paper for publication after minor revision addressing the issues listed below:

ANSWERS:

We thank the reviewer for his/her comments and evaluation of our work.

Below we will reply comment by comment.

1. In Figures S1 C and S3, the ordinate variables should be followed only by units, e.g., “Absorbance, a.u.”, with explanation in Figure caption that 2 mm pathway cells were used. Alternatively, the correct mathematical expression: “A/5” (rather than “Absorption/5”), followed by “a.u.” units could be used.

The figures are corrected. We did write “a.u.” since absorption does not have units. Since absorption is logI/I0, it is a number where I is light intensity at a certain wavelength.

2. Figure captions, in main text and supporting information, should contain all important experimental conditions (e.g., concentrations of analytes, solutions, pH). In Figure 2 caption, the name of compound should be added.

Figure captions are corrected.

3. It seems that in addition to the radiosensitizing effects of AuNPs on the oxidation processes of certain small biomolecules, investigated by the Authors, AuNPs may also induce significant damage to DNA in the presence of ionizing radiation. For instance, it has been found that premutagenic changes in DNA are made by *OH and Cu(I)-hydroperoxyl complex radicals generated by Fenton-type mechanisms in the presence of Fe2+ or Cu2+ ion and catechol moieties (Mutation Research - Fundamental and Molecular Mechanisms of Mutagenesis, 2012, 735, 1– 11). Any comment on this subject would benefit the general Readership.

Thank you for the proposed paper. However, we would like to use this citation for our next paper to discuss AuNP catalytic reactions involving oxygen. As you can see in this paper, all solutions were deoxygenated.

4. I have not found any typographical or English errors.

Reviewer 3 Report

In this manuscript, Mostafavi et al reported the effect of AuNPs on selective oxidation of transient organic radicals in the 2-propanol oxidation and acetanilide hydroxylation in aqueous solutions. This paper can be published in Nanomaterials after my comments are addressed:

  1. Large scale of TEM images of Au NPs should be provided.
  2. The detailed calculation method for the G (Figure 1 and 3) should be provided.
  3. Can the authors provide more discussion that why Au NPs can selectively oxidize the transient organic radicals? What is intrinsic role of Au NPs?
  4. Is enhanced effect exclusively for Au NPs?
  5. The full name of ERS should be provided.

Author Response

In this manuscript, Mostafavi et al reported the effect of AuNPs on the selective oxidation of transient organic radicals in the 2-propanol oxidation and acetanilide hydroxylation in aqueous solutions. This paper can be published in Nanomaterials after my comments are addressed:

ANSWERS:

We thank the reviewer for his/her comments.

Below we will reply comment by comment.

1. Large-scale of TEM images of Au NPs should be provided.

We have included a large-scale TEM image.

2. The detailed calculation method for the G (Figure 1 and 3) should be provided.

For figures 1 and 3 ”Radiolytic yield (G) of acetone is determined from a linear slope.” (lines 176 and 217).

3. Can the authors provide more discussion that why Au NPs can selectively oxidize the transient organic radicals? What is the intrinsic role of Au NPs?

See response to question 4.

4. Is enhanced effect exclusively for AuNPs?

Questions 3 and 4 are discussed in the last paragraph of section 3.1. Where we also refer to other papers discussing these questions.

Lines 176-182: “Conducting nanoparticles can accumulate a negative charge by uptaking unpaired electrons from α-hydroxyisopropyl radical [34], that is supported by the concept of “flouting electrocatalysis” [35].”

5. The full name of ERS should be provided.

ERS was changed to the full name of the method “studied by electron paramagnetic resonance spectroscopy” (line 36).

Round 2

Reviewer 1 Report

The Authors revised their manuscript accordingly. I suggest the Authors to correct two small problems prior publication:

1. The Authors wrote (former question 10) “Indeed, acetone concentration in a non-irradiated solution (Figure S3) is 18.2 mM...”. I agree that the initial concentration does not matter when monitoring acetone concentration increase. However, this does not solve the problem with the plot Fig S3: the absorption is not consistent with the concentration. Please add a note, perhaps in Fig. S3 caption, that the acetone concentration used exactly in that test was 18.2 mM, but not 20 mM. This would avoid confuse if someone decide to use these UV/Vis data as a reference ones.

2. The Authors replied “The stationary concentration of OH radicals and R•OH is low (10-9M) compared to the concentration of 2-propanol (100mM). Therefore, the reaction between OH radicals and 2-propanol is much more favorable than between OH radicals and R•OH”. This is an acceptable explanation. However, I strongly recommend the Authors to improve Scheme I and clearly indicate the disproportionation pathway and the AuNPs oxidation-reduction catalytic cycle. This would make the paper more informative to readers.

Author Response

Thank you for your comment,

1. The Authors wrote (former question 10) “Indeed, acetone concentration in a non-irradiated solution (Figure S3) is 18.2 mM...”. I agree that the initial concentration does not matter when monitoring acetone concentration increase. However, this does not solve the problem with the plot Fig S3: the absorption is not consistent with the concentration. Please add a note, perhaps in Fig. S3 caption, that the acetone concentration used exactly in that test was 18.2 mM, but not 20 mM. This would avoid confuse if someone decide to use these UV/Vis data as a reference ones.

According to your suggestion, we have changed the caption of Fig. S3 and Fig. 1.

2. The Authors replied “The stationary concentration of OH radicals and R•OH is low (10-9M) compared to the concentration of 2-propanol (100mM). Therefore, the reaction between OH radicals and 2-propanol is much more favorable than between OH radicals and R•OH”. This is an acceptable explanation. However, I strongly recommend the Authors to improve Scheme I and clearly indicate the disproportionation pathway and the AuNPs oxidation-reduction catalytic cycle. This would make the paper more informative to readers.

Herein, we would like to point out, that this scheme of acetone formation in an irradiated solution of 2-propanol is known (please see, [35] line 157). This scheme is only used to show the reactions occurring in the solution, it is not a scheme of the processes the same as Scheme 2.